# Learn from Downstream and Be Yourself in Multimodal Large Language Models Fine-Tuning

Wenke Huang [* 1]  Jian Liang [* 1]  Zekun Shi [1]  Didi Zhu [2]  Guancheng Wan [1]  He Li [1]  Bo Du [† 1]  Dacheng Tao [3]
Mang Ye [† 1]

## Abstract

Multimodal Large Language Model (MLLM) has demonstrated strong generalization capabilities across diverse distributions and tasks, largely due to extensive pre-training datasets. Fine-tuning MLLM has become a common practice to improve performance on specific downstream tasks. However, during fine-tuning, MLLM often faces the risk of forgetting knowledge acquired during pre-training, which can result in a decline in generalization abilities. To balance the trade-off between generalization and specialization, we propose measuring the parameter importance for both pre-trained and fine-tuning distributions, based on frozen pre-trained weight magnitude and accumulated fine-tuning gradient values. We further apply an importance-aware weight allocation strategy, selectively updating relatively important parameters for downstream tasks. We conduct empirical evaluations on both image captioning and visual question-answering tasks using various MLLM architectures. The comprehensive experimental analysis demonstrates the effectiveness of the proposed solution, highlighting the efficiency of the crucial modules in enhancing downstream specialization performance while mitigating generalization degradation in MLLM Fine-Tuning.

## 1. Introduction

Recent years have witnessed remarkable progress in Multimodal Large Language Model (MLLM), which have

[*]Equal contribution [1]National Engineering Research Center for Multimedia Software, School of Computer Science, Wuhan University, Wuhan, China [2]Department of Computer Science and Technology, Zhejiang University, Hangzhou, China [3]Nanyang Technological University, Singapore. Correspondence to: Mang Ye <yemang@whu.edu.cn>, Bo Du <dubo@whu.edu.cn>.

*Proceedings of the 42nd International Conference on Machine Learning*, Vancouver, Canada. PMLR 267, 2025. Copyright 2025 by the author(s).

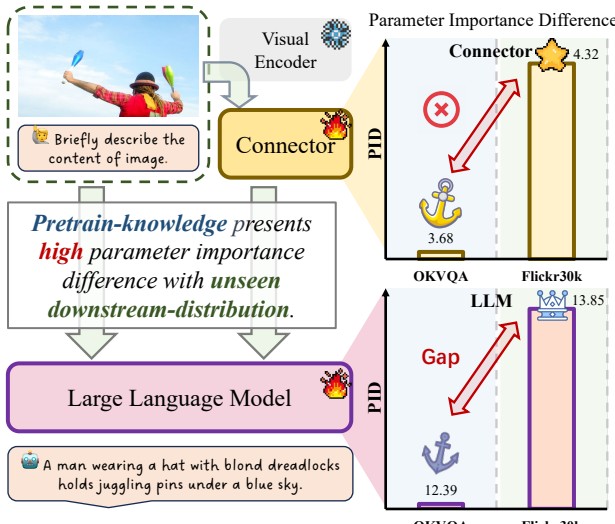

Figure 1: **Background and Motivation**. Fine-tuning Multimodal Large Language Model (MLLM) on downstream tasks typically involves training (🔥) the connector and LLM modules, and freezing (❄) the visual encoder. We reveal a **higher** parameter importance difference (**PID**) on **unseen downstream distributions**, *e.g.*, Flickr30k, compared to **seen upstream distribution**, *e.g.*, OKVQA. PID = $cos(|w^*|, |g|)^{-2}$. We utilize the absolute value of the pre-trained weight $|w^*|$ and fine-tuning gradients $|g|$ to represent the upstream and downstream parameter importance.

demonstrated impressive competency in various vision-understanding tasks (Liu et al., 2023b; Dai et al., 2023; Lin et al., 2023a; Liu et al., 2023a). MLLM generally follows the paradigm to fuse the pre-trained vision encoder (Radford et al., 2021; Dosovitskiy et al., 2021) into the representation space of the Large Language Models (LLM), *e.g.*, LLaMA (Touvron et al., 2023) and Vicuna (Chiang et al., 2023), via the connector module (Dai et al., 2023; Liu et al., 2023b; Luo et al., 2024). Considering that Multimodal Large Language Model is optimized on huge-scale and various-type multimodality instruction-following datasets (Lin et al., 2014; Singh et al., 2019; Mishra et al., 2019), it brings powerful generalization ability on different related tasks. Despite this, MLLM still performs poorly on downstream datasets (Lin et al., 2023b; Luo et al., 2023; Zhai et al., 2024; Tang et al., 2024; Lu et al., 2024).

The common practice is to fine-tune foundation models

on specific tasks to enhance task performance or align the model behavior with human expectations (Zhou et al., 2024; Han et al., 2024). Specifically, existing solutions normally freeze the visual encoder, focusing solely on connector layers and the LLM component (Su et al., 2023; Cha et al., 2024; Lin et al., 2024). Thus, during the fine-tuning stage, the MLLM gains specialization ability to achieve exceptional performance on the fine-tuning task. However, since the small fine-tuning dataset does not have sufficient coverage of the distribution as well as tasks, the fine-tuned model can potentially lose its generality which is acquired through pre-training stage. The effect of deteriorating the model previous generic knowledge upon new learning is a well-documented challenge, referred as catastrophic forgetting (Ratcliff, 1990; McCloskey & Cohen, 1989; French, 1999; Luo et al., 2023; Zhai et al., 2024). Consequently, a pivotal question raises: *How to enhance the specialization ability on target tasks while maintaining generalization knowledge for Multimodal Large Language Model Fine-Tuning?*

In our work, we propose a simple yet effective method, **Sp**ecialization via **I**mportance **D**iscrepancy **E**valuation for **R**efinement, abbreviated as SPIDER (🕷) to simultaneously accommodate task-oriented properties and maintain task-generic knowledge to alleviate the catastrophic forgetting phenomenon. Own to the over-parameterized characteristics of the deep neural network (Guo et al., 2017; Lakshminarayanan et al., 2017), we notice that not all parameters contribute equally to fit the target distribution, which has confirmed soundness in the relative researches (Li et al., 2017; Frankle & Carbin, 2019; Liu et al., 2019; Sung et al., 2021; Yin et al., 2023; Sun et al., 2023b). Consequently, we assert that pre-training and fine-tuning distributions exhibit distinct parameter importance degrees for representing generic and specialized knowledge. In line with this, we are motivated to ***selectively update relatively important parameters for the downstream task while preserving the remaining for generalization and specialization ability***.

Driven by this motivation, the above problem becomes more fundamental: **I)**: *How to measure parameter importance for generic and specialized knowledge*? **II)**: *How to selectively maintain and optimize during downstream learning*. To respond to question **I)**, we introduce the Importance Discrepancy Measurement (IDM) to quantify the parameter importance degree towards the generalized and specialized knowledge. With respect to the prior generic information, we take inspiration from (Han et al., 2015; Frankle & Carbin, 2019; Sun et al., 2023b) and argue that *weight magnitude* positive correlates with the prediction tendency. Thus, we adopt the MLLM **pretrained weight magnitude** to measure the parameter importance towards the generalization attitude. Towards the posterior specialized behavior, we utilize **current gradient norm** to pinpoint regions that learn crucial downstream knowledge. The rationale behind this

is that *gradients trajectories* directly provides intensity information of the learning signal imposed on each parameter element for optimization objective (Fisher, 1922; Pascanu & Bengio, 2013; Kirkpatrick et al., 2017; Lee et al., 2019; Mirzadeh et al., 2020; Sanh et al., 2020; Song et al., 2024). By comparing the parameter importance rank, we can differentiate between generic and task-specific parameters, As illustrated in Fig. 1. tuning on the unseen fine-tuning datasets, *e.g*., Flickr30k, exhibits a more pronounced parameter importance discrepancy than the seen OKVQA distributions. This observation further supports our motivation to mitigate catastrophic forgetting in MLLM by considering parameter importance discrepancy. Driven by question **II)**, we then propose the Importance Selection Mask (ISM) to selectively consolidate or optimize candidate parameters for the target distribution. Specifically, during the network backward pass, we identify and consolidate parameters that exhibit relatively higher importance for general task knowledge, while optimizing the remaining elements to enhance task-specific performance. For thorough examination, we conduct experiments on two representative MLLM: VILA (Lin et al., 2023a) and LLaVA (Liu et al., 2023b;a). We fine-tune on two major tasks image-captioning and visual question answering and evaluate the generic knowledge on the pre-trained seen datasets (Goyal et al., 2017; Marino et al., 2019; Hudson & Manning, 2019; Singh et al., 2019). The main contributions are summarized as follows:

- We focus on addressing the catastrophic forgetting problem in fine-tuning Multimodal Large Language Model (MLLM) for downstream tasks. We reveal that due to the distribution shift between upstream and downstream patterns, parameters exhibit varying degrees of importance.

- We introduce SPIDER, which measures generalization and specialization based on the behavior of frozen weights and updating gradients. Our method identifies relatively important elements for the downstream task and conduct critical-aware weight allocation on candidate parameters. By selecting and ranking these elements, our approach offers a novel solution to effectively tackle the generalization and specialization dilemma in MLLM.

- We conduct a comprehensive analysis on four downstream datasets: Flickr30k (Young et al., 2014), COCO-Capation (Lin et al., 2014), ScienceQA (Lu et al., 2022), and IconQA (Lu et al., 2021), using VILA (Lin et al., 2023a) and LLaVA (Liu et al., 2023b). Along with a series of ablation studies, the promising results empirically validate the effectiveness of SPIDER in improving fine-tuning performance and mitigating generalization forgetting.

## 2. Related Works

### 2.1. Multimodal Large Language Models

With the impressive success of Large Language Models (LLM), such as GPT (Radford et al., 2019; Brown et al.,

2020; OpenAI, 2023), LLaMA (Touvron et al., 2023), Vicuna (Chiang et al., 2023), PaLM (Chowdhery et al., 2022; Anil et al., 2023), growing interest has been aroused in building end-to-end Multimodal Large Language Model (MLLM), *e.g.*, Flamingo (Alayrac et al., 2022), BLIP-2 (Li et al., 2022a; 2023a), InstructBLIP (Dai et al., 2023), QWen-VL (Bai et al., 2023), LLaVA (Liu et al., 2023b;a; ?), VILA (Lin et al., 2023a). Existing MLLM solutions normally follow to utilize the visual extractor (Radford et al., 2021; Dosovitskiy et al., 2021) to encode visual features and utilize the connector module to project visual tokens into word embedding space of the LLM, *i.e.*, treating visual input as the foreign language (Wang et al., 2023c). Then, the visual and textual tokens are concatenated and fed into the LLM. The LLM is used to accomplish various vision-language tasks in an auto-regressive manner. For example, the famous MLLM work, LLaVA (Liu et al., 2023b) adopts a linear projection layer to connect the visual encoder and the LLM (Chiang et al., 2023; Touvron et al., 2023). Despite their effectiveness, existing works primarily emphasize the generalization ability across various tasks, resulting in the constrained performance on specific downstream target tasks. Therefore, it is an intuitive solution to fine-tune the MLLM in order to enhance the particular task performance.

## 2.2. Catastrophic Forgetting in Multimodal Large Language Model Fine-Tuning

Commonly optimized on downstream tasks (De Boer et al., 2005), deep neural network is empirically proved to suffer from the *catastrophic forgetting* problem (Ratcliff, 1990; McCloskey & Cohen, 1989; French, 1999; Luo et al., 2023; Kirkpatrick et al., 2017; Zhai et al., 2024), a significant issue where models forget previously learned information when exposed to new data. In the context of MLLM, this results in catastrophic forgetting of generic knowledge, which severely impairs the model transferability across previously learned datasets. Therefore, balancing the ability to fit downstream tasks while maintaining generalization becomes a crucial challenge for Multimodal Large Language Model. Existing methods could be roughly divided into four categories (Lin et al., 2023b; Han et al., 2024; Xin et al., 2024; Huang et al., 2025). **i)** *Additive Parameter* Learning (Houlsby et al., 2019; Yi-Lin Sung, 2022; Lester et al., 2021; Liu et al., 2022; Zhou et al., 2022b;a; Zhang et al., 2022) primarily focus on strategically incorporating additional trainable parameters within the architecture. For example, adapter (Houlsby et al., 2019; Zhang et al., 2022; Li et al., 2023b; Gao et al., 2023; Sun et al., 2023a) typically consist of multi-layer perceptions and residual connections (He et al., 2016) that combine pre-trained features with updated ones. Additionally, prompt (Lester et al., 2021; Zhou et al., 2022b;a; Zang et al., 2022; Shu et al., 2022; Wang et al., 2022) directly appends adjustable vectors to the

Table 1: **Limitation** for different Anti-Forgetting MLLM methods: Additive Parameter Learning (Add.), Reparameterization Tuning (Repara.), Regularization based Optimization (Reg.), and Partial-based Updating (Part.). Refer to Sec. 2.2 for details.

| Limitation | Add. | Repara. | Reg. | Part. | Ours |
|---|---|---|---|---|---|
| **Specify** Architecture | ✓ | ✓ | | | ✗ |
| **Modify** Optimization | | | ✓ | | ✗ |
| **Require** Hyper-Parameter | | | ✓ | ✓ | ✗ |

input sequence. **ii)** *Reparameterization Tuning* (Hu et al., 2022; Zhang et al., 2023; Wang et al., 2023a; Hao et al., 2024; Liu et al., 2024; Bi et al., 2025; Liang et al., 2025) also introduce new learnable parameters during the training stage, which are then integrated into the original MLLM through reparameterization during inference. For instance, LoRA (Hu et al., 2022) assumes that the changes in linear model weights follow a low-rank behavior. Despite the certain advantages, these two research streams introduce additional parameters into the pre-trained model and disrupt the original architecture, leading to increased computational costs and presents restricted architecture compatibility. **iii)** *Regularization-based Optimization* (Kirkpatrick et al., 2017; Zenke et al., 2017; Xuhong et al., 2018; Ritter et al., 2018; Buzzega et al., 2020; Li et al., 2020; Panigrahi et al., 2023) introduce the loss constraints to preserve the previously learned knowledge. Several studies add regularization terms to the loss functions to penalize parameter changes and mitigate catastrophic forgetting. However, aforementioned solutions require to modify the loss function and thus conflict with personalized fine-tuning loss design. **iv)** *Partial-based Updating* (Li et al., 2022b; Ansell et al., 2022; Li et al., 2023c; Yu et al., 2024; Zhang et al., 2024c;b; Zhu et al., 2024; Lu et al., 2024) focuses on modifying a subset of downstream-relevant parameters, making it architecture-agnostic and orthogonal to the downstream loss objective. For instance, GPS (Zhang et al., 2024c) and SPU (Zhang et al., 2024b) perform sparse updates based on gradient signals, while DARE (Yu et al., 2024) and Tailor (Zhu et al., 2024) operate on delta parameters. However, previous methods struggle to retain generic knowledge and their performance is highly sensitive to predefined selection thresholds. In our research, recognizing the distinct characteristics of deep neural networks, we argue that parameters exhibit differing importance distributions between pre-training and fine-tuning phases. Therefore, we measure parameter importance in a self-driven manner, selectively updating those with relatively higher importance for downstream tasks while preserving the generalization capability.

## 3. Methodology

### 3.1. Preliminary

Given the Multimodal Large Language Model (MLLM) architecture, the MLLM model ($\theta$) typically includes three parts: visual encoder $f$, *e.g.*, ViT (Dosovitskiy et al., 2021),

LLM ($g$), *e.g.*, Vicuna (Chiang et al., 2023) and LLaMA (Touvron et al., 2023), and the connector module $\varphi$ (Liu et al., 2023b; Dai et al., 2023; Liu et al., 2023a; Lin et al., 2023a). For a query instance, the input consists of both a visual image $x^v$ and a textual instruction $x^t$. The corresponding label is a language response $y$. First, we extract the visual features $z^v = f(x^v)$, and then apply the trainable projection $\varphi$ to convert $z^v$ into language embedding tokens, $h^v = \varphi \cdot z^v$. And textual token as $h^t = \text{Tokenize}(x^t)$. Next, we combine both visual and textual tokens and pass them into the LLM module $g$ to generate the language output $\hat{y} = g([h^v, h^t])$. Following previous MLLM fine-tuning works and benchmarks (Zhou et al., 2024; Zhu et al., 2024), we select and fine-tune partial trainable parameter module $w$ from the MLLM model to adapt to the downstream task $\mathcal{T}$ with distribution $(D^{\mathcal{T}})$. Normally, learnable modules are connector module ($\varphi$) and candidate LLM ($g$) blocks as $w = \{\varphi, g\}$. This MLLM optimization follows:

$$\arg\min_w \mathbb{E}_{(x^v, x^t, y) \in \mathcal{D}^\mathcal{T}} \mathcal{L}\left(g([\varphi(h^v), h^t]), y\right). \quad (1)$$

### 3.2. Specialization via Importance Discrepancy Evaluation for Refinement

To enhance downstream efficiency while preserving generic knowledge in MLLM, we assess parameter importance across pre-training and fine-tuning distributions, selectively updating downstream critical elements, including two components: Importance Discrepancy Measurement (IDM Sec. 3.2.1) for ranking parameter importance, and Importance Selection Mask (ISM Sec. 3.2.2) for selective updates.

#### 3.2.1. IMPORTANCE DISCREPANCY MEASUREMENT

**Importance for Generalization Knowledge**. Generic knowledge embedded in MLLM provides bases for strong performance in various domains and quick transfer to different tasks; when directly fine-tuning on newly received tasks with no regard to preserving its pre-existing, MLLM faces the catastrophic forgetting on the generalization ability. Thus, with respect to the generalization knowledge, we take inspiration from the magnitude pruning (Han et al., 2015) and weight magnitude represents how much the parameter contributes to the model prediction (Frankle & Carbin, 2019). Thus, in our work, we directly utilize the weight magnitude (Han et al., 2015; Frankle & Carbin, 2019; Sun et al., 2023b) for pre-trained parameters $w^*$ to rank the generalization parameter importance $\mathcal{I}$ as the following formulation.

$$\mathcal{I}[v] = |w^*[v]| \qquad \text{Absolute,} \quad (2a)$$

$$\mathcal{I}[v] = \frac{\mathcal{I}[v] - \text{Mean}(\mathcal{I})}{\text{Std}(\mathcal{I})} \qquad \text{Normalization,} \quad (2b)$$

$$\mathcal{I}[v] = \frac{1}{1 + e^{-\mathcal{I}[v]}} \qquad \text{Rescale.} \quad (2c)$$

In this context, the notation $[v]$ reresents the $n^{th}$ component value of a given tensor vector. The role of the above form is threefold. Eq. (2a) computes the weight magnitude, and Eq. (2b) is applied to eliminate the effect of dimensional

analysis. We further rescale into the bounded range for comparison via Eq. (2c). Thus, $\mathcal{I}[v]$ is within $(0, 1)$

**Importance for Specialization Knowledge**. With respect to fine-tuning the downstream task, we aim to identify which parameters are more relevant to the specific task at hand. We argue that the gradient signal acts an effective evaluation metric as the following formulation:

$$\delta[v] = \frac{\partial \mathcal{L}\left(g([\varphi(h^v), h^t]), y\right)}{\partial w[v]}, \quad (3)$$

where $\delta[v]$ denotes the gradient of the loss function with respect to the parameter $w[v]$, evaluated at the query sample. The intuition is that parameters with larger gradient values correspond to directions where the loss function changes most rapidly, facilitating efficient gradient descent during fine-tuning. Thus, we derive the specialization parameter importance $\mathcal{G}$. The formulation is quantified as follows:

$$\mathcal{G}[v] = \text{Norm}(|\delta[v]|) \in (-\infty, \infty),$$
$$\mathcal{G}[v] = \text{Sigmoid}(\mathcal{G}[v]) \in (0, 1). \quad (4)$$

Notably, due to the stochastic nature of sampling, $\delta[v]$ exhibits instability in reflecting parameter importance, thereby introducing considerable uncertainty in estimating specialized knowledge. To address this issue, we draw inspiration from the momentum mechanism (He et al., 2020; Chen et al., 2020; Chen* et al., 2021) and iteratively accumulate sample gradients using a momentum update with a coefficient of 0.85 to reduce the impact of this uncertainty.

Therefore, we evaluate parameter importance for both generalization and specialization by utilizing pre-training weights and fine-tuning gradient information. To ensure a balanced assessment, we apply the consistent normalization and rescaling methods to these two metrics.

#### 3.2.2. IMPORTANCE SELECTION MASK

After localizing the parameter importance for both generalization ($\mathcal{I}$) and specialized knowledge ($\mathcal{G}$) during the fine-tuning stage, we only update the selected parameters while keeping the remaining pre-trained model parameters frozen. Thus, the straightforward approach is to treat the relatively important elements for the downstream task as candidate parameters for updates. Thus, we define the updating mask $\boldsymbol{M}$ as the following formulation:

$$\boldsymbol{M}[v] = \begin{cases} 1, & \mathcal{G}[v] > \mathcal{I}[v], \\ 0, & \text{else.} \end{cases} \quad (5)$$

When $\boldsymbol{M}[v] = 1$, the query parameter is selected as updating candidate. We denote the current MLLM model as $w$. We utilize the frozen pre-trained parameter $w^*$ to reweight the current model, thereby restoring the original pre-trained knowledge conditions as follows:

$$w = w \odot \boldsymbol{M} + w^* \odot (1 - \boldsymbol{M}). \quad (6)$$

However, this operation introduces no variance in the candidate parameter updates. Moreover, due to its normalization property, the aforementioned solution can be seen as mask-

ing fifty percent of the parameter updates, which still results in the degradation of generalization performance. We argue that for the selected parameters, assigning higher weights to those exhibiting a greater discrepancy in importance, and conversely lower weights to less significant parameters. Thus, we propose the Importance Selection Mask (ISM) to reconstruct the aggregation weight in Eq. (5) as:

$$\boldsymbol{M}[v] = \begin{cases} \frac{\mathcal{G}[v]}{\mathcal{G}[v]+\mathcal{I}[v]}, & \mathcal{G}[v] > \mathcal{I}[v], \\ 0, & \text{else}. \end{cases} \quad (7)$$

Furthermore, we rescale the aggregation weights based on the mean behavior of the selected elements, while restricting the upper bound to 1. This can be considered as the following rescale operation strategy:

$$\boldsymbol{M}[v] = \begin{cases} \min(1, \frac{\boldsymbol{M}[v]}{\text{Mean}(\boldsymbol{M}[\boldsymbol{M}\neq 0])}), & \mathcal{G}[v] > \mathcal{I}[v], \\ 0, & \text{else}. \end{cases} \quad (8)$$

Based on the above Importance Selection Mask (ISM), we rewrite the Eq. (6) to update the current model and plot the algorithm description in Algorithm 1 and Fig. 2.

### 3.3. Discussion and Limitation

**Related Parameter Signal Investigations.** Generally speaking, parameter signals could be revealed in two aspects: magnitude (Han et al., 2015; Frankle & Carbin, 2019; Sun et al., 2023b) and gradient (Fisher, 1922; Pascanu & Bengio, 2013; Kirkpatrick et al., 2017; Lee et al., 2019; Mirzadeh et al., 2020; Sanh et al., 2020). The weight magnitude represents how much the parameter contributes to the prediction. The gradient reveals the information intensity during optimization. Thus, magnitude and gradient acts as parameter importance metrics to select target elements, which has incurred huge research interest in broad fields, such as network pruning (Han et al., 2015; Frankle & Carbin, 2019; Zhang et al., 2021; Li et al., 2022b), domain generalization (Rame et al., 2022; Wang et al., 2023b; Zhu et al., 2023a), federated learning (Sung et al., 2021; Matena & Raffel, 2022), and malicious defense (Han et al., 2023; Zhu et al., 2023c; Huang et al., 2023; Zhu et al., 2023b; Huang et al., 2024b). Existing explorations focus on training a network from scratch and face no requirement to preserve the previously learned knowledge, thus entangling the magnitude and gradient information to select the crucial elements for the target task. However, pre-trained Multimodal Large Language Model (MLLM) models have inherent generalization knowledge, as evidenced by the capacity to execute diverse tasks without fine-tuning (Liu et al., 2023b;a; Zhang et al., 2024a). Thus, maintaining the generalization ability and enhancing the downstream specialization ability during the fine-tuning stage acts as a crucial task for MLLM. In our work, we utilize the pre-trained parameter magnitude ($\mathcal{I}$ in Eq. (2a)) and optimizing parameter gradient ($\delta$ in Eq. (3)) to respectively reveal the parameter importance metrics for the generalization and specialization abilities. We select relative downstream-kernel elements to balance the generalization

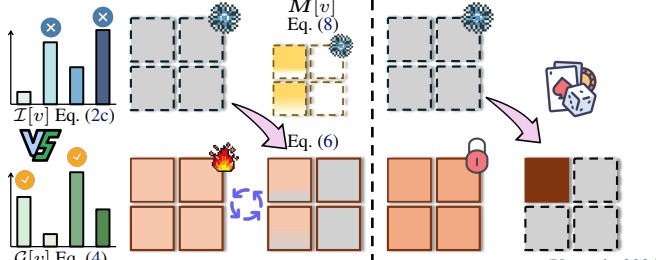

Figure 2: **Conceptual Comparison.** (a) SPIDER iteratively measures the parameter importance discrepancy to construct the update mask which protects generation and squeezes specialization information on the selected elements. (b) DARE combines the learned elements with the pre-trained and further rescale the candidate ones. 🧊 means frozen pre-trained elements. 🔥 denotes current learning parameters. 🔒 represents completed learned ones.

and specialization ability during the fine-tuning process.

**Relative Model Merging Works.** Large Model Merging is a promising paradigm that integrates task-specific models into a unified model capable of simultaneously handling diverse downstream tasks (Zheng et al., 2025). Existing methods typically represent each task expert using downstream-optimized models and rely on various importance metrics to select specific elements for merging (Tang et al., 2024; Zhu et al., 2024; Yu et al., 2024; Lu et al., 2024; Du et al., 2024; Huang et al., 2024a; Zhu et al., 2025). However, post-merging operations often disrupt the originally optimized parameter space, limiting the specialized capabilities for individual tasks. In our work, we select relative specialization importance elements during the training process, which is compatible with integrates into the training trajectory and squeeze specialized capabilities into target parameter space.

**Concept Difference.** Existing methods to mitigate MLLM forgetting, such as DARE (Yu et al., 2024) and Tailor (Zhu et al., 2024), primarily focus on selectively updating and rescaling optimized parameters using random selection and the Hessian matrix (Fisher, 1922; Sung et al., 2021; Tang et al., 2024; Frantar & Alistarh, 2023). However, these post-combination operations can conflict with optimization strategies that aim to adjust all trainable elements for downstream performance and is sensitive with the changing scale (Ilharco et al., 2023; Yu et al., 2024). Our approach evaluates parameter importance for both generalization and specialization objectives during the tuning stage. This enables us to selectively update parameters relevant to downstream tasks while preserving others, effectively performing an *information extrusion* (Frankle & Carbin, 2019; Chen et al., 2021; Zhang et al., 2021; Bai et al., 2022) to reduce conflicts between pre-training and fine-tuning knowledge. We illustrate the concept difference in Fig. 2.

**Limitation.** SPIDER leverages both previously pre-trained and current fine-tuning knowledge to select relevant downstream important elements and re-weight the candidate pa-

Table 2: **Computation Complexity, Learnable Ratio and Performance Comparison**. Accuracies are derived from Flickr30k based on VILA. $\mathcal{O}$ denotes the complexity degree. Refer to Sec. 3.3.

| Metrics | Full FT | L2-Reg | Grafting | Half FT | DARE | Tailor | SPIDER |
|---|---|---|---|---|---|---|---|
| Complexity | $\mathcal{O}(\lvert M\rvert)$ | $\mathcal{O}(2\times\lvert M\rvert)$ | $\mathcal{O}(2\times\lvert M\rvert)$ | $\mathcal{O}(\lvert M\rvert)$ | $\mathcal{O}(\lvert M\rvert)$ | $\mathcal{O}(\lvert M\rvert)$ | $\mathcal{O}(3\times\lvert M\rvert)$ |
| Ratio | 100% | 100% | 100 % | 50 % | 10% | 10 % | 50% |
| Performance | 55.16 | 48.33 | 49.58 | 59.52 | 53.66 | 54.56 | 66.61 |

rameters, thereby ensuring generalization while pressing the fine-tuning optimization pathway. However, ours fails in certain circumstances. (i) We rank parameter importance based on the pre-trained weights and the current gradient matrix, which incurs additional memory usage. However, this increase is linear relative to the scale of learnable parameters, with a resource complexity of $\mathcal{O}(3\times\lvert M\rvert)$. We further plot the comparison in Tab. 2. (ii) Our method evaluates parameter importance by jointly considering pre-training and fine-tuning distributions, enabling the selection of task-relevant parameters that balance generalization and specialization. When the downstream task closely aligns with the upstream distribution, only minimal updates are required. In such cases, small distribution shifts result in an acceptable level of generalization loss, thereby effectively managing the trade-off between preserving broad capabilities and adapting to specific tasks.

## 4. Experiments

### 4.1. Experimental Setup

**Architecture and Datasets**. Adhering to the Multimodal Large Language Model paradigm, we evaluate the effectiveness of our methods using two popular models as the foundations for our experiments: LLaVA (Liu et al., 2023b) and VILA (Lin et al., 2023a). We categorize the datasets into two groups: pre-training (seen) and fine-tuning (unseen) datasets to respectively measure the generalization and specialization ability. The pre-training datasets consist of those used in the training process; accordingly, we assess the learned generalization ability on OKVQA (Marino et al., 2019), TextVQA (Singh et al., 2019), GQA (Hudson & Manning, 2019), and OCRVQA (Mishra et al., 2019). For fine-tuning tasks, we consider four downstream datasets: Flickr30k (Young et al., 2014), COCO-Capation (Lin et al., 2014), IconQA (Lu et al., 2021), ScienceQA (Lu et al., 2022)[1], which respectively associate with image caption and visual reasoning views. To be precise, OKVQA, TextVQA , GQA, and OCRVQA are obviously mentioned as the training datasets in the pre-training stage, making them appropriate benchmarks to evaluate multimodal large language models (MLLMs) generalization across diverse tasks. OKVQA examines external knowledge and common-sense reasoning, TextVQA and OCRVQA test understanding of embedded textual information, and GQA assesses compositional and logical reasoning. Together, these datasets comprehensively

[1]https://huggingface.co/datasets/BAAI/DataOptim

evaluate MLLMs generalization across different reasoning types and practical scenarios. We follow (Zhou et al., 2024) resource setting and randomly sample $10k$ samples from the training set of each dataset.

**Counterparts**. We focus on exploring model-agnostic MLLM fine-tuning methods and mainly compare with the Regularization-based Optimization and Partial-based Updating solutions as follows: Full Fine-Tuning (Full FT) [arXiv'05] (De Boer et al., 2005), L2-Regularization (L2-Reg) [PNAS'17] (Kirkpatrick et al., 2017), Grafting [ICML'23] (Panigrahi et al., 2023), Half Fine-Tuning (Half FT) [arXiv'24] (Hui et al., 2024), DARE [ICML'24] (Yu et al., 2024), and Tailor [ICML'24] (Zhu et al., 2024).

**Implementation Details**. We follow the official codebase[2,3] to conduct the fine-tuning procedure. The learning rate $lr$ in LLaVA (Liu et al., 2023b) is $2e-4$ for LLM and $2e-5$ for visual projector. For VILA (Lin et al., 2023a), we uniformly set the learning rate to $1e-4$. The training epoch is $E = 5$. The training batch size $B$ set 16. The fine-tuning block for LLM is the *last $L = 2$ layers*. All experiments are conducted on 8 NVIDIA 4090 GPUs, each with 24GB memory. Due to limited computation resources, we utilize LLaVA-1.5-7B for LLaVA and VILA1.5-3B for VILA.

**Evaluation Metrics**. To evaluate the performance of Multimodal Large Language Model (MLLM) in both generalization and specialization aspects, we consider two key metrics: Source Performance ($\mathcal{A}^{\mathcal{S}}$) and Target Performance ($\mathcal{A}^{\mathcal{T}}$). Let $\mathcal{U} = \{\mathcal{U}_i\}_{i=1}^{\lvert\mathcal{U}\rvert}$ represent the set of pre-training datasets and $\mathcal{T}$ denote the fine-tuning target dataset. Thus, we derive the following evalation metrics forms:

$$\mathcal{A}^{\mathcal{S}} = \frac{1}{\lvert\mathcal{U}\rvert}\sum_{i}^{\lvert\mathcal{U}\rvert}\text{Acc.}(\mathcal{U}_i), \quad \mathcal{A}^{\mathcal{T}} = \text{Acc.}(\mathcal{T}). \qquad (9)$$

Acc. denotes the accuracy metric. We use the CIDEr metric to evaluate performance on the Flickr30k (Young et al., 2014) and COCO-Capation (Lin et al., 2014) datasets. For simplicity, we apply the same notation throughout. To evaluate effectiveness in mitigating catastrophic forgetting in MLLM, we use the H-Average metric ($\mathcal{H}$) and O-Average metric ($\mathcal{O}$) (Zhu et al., 2024). The H-Average and O-Average metrics measure the harmonic and arithmetic mean of generalization ($\mathcal{A}^{\mathcal{S}}$) and specialization ($\mathcal{A}^{\mathcal{T}}$):

$$\mathcal{H} = \frac{2\times\mathcal{A}^{\mathcal{S}}\times\mathcal{A}^{\mathcal{T}}}{\mathcal{A}^{\mathcal{S}}+\mathcal{A}^{\mathcal{T}}}, \quad \mathcal{O} = \frac{\mathcal{A}^{\mathcal{S}}+\mathcal{A}^{\mathcal{T}}}{2}. \qquad (10)$$

### 4.2. Diagnostic Analysis

We ablation on Flickr30k and IconQA for in-depth analysis.

**Candidate Parameter Selection Metrics**. Selecting candidate parameters plays a crucial role in mitigating catastrophic forgetting of general knowledge while enhancing

[2]https://github.com/haotian-liu/LLaVA
[3]https://github.com/NVlabs/VILA

Table 3: **Ablation Analysis for Candidate Parameters Selection**. The $\text{Rand}_\gamma$ denotes randomly selects elements with $\gamma$ ratio. $\mathcal{I}_\gamma$ and $\mathcal{G}_\gamma$ respectively denotes choose $\gamma$ proportion of elements via magnitude and gradient. $\gamma$ is set as $50\%$. Please see Sec. 4.2.

| Metric | Flickr30k | | | COCO-Capation | | |
|---|---|---|---|---|---|---|
| | $\mathcal{A}^\mathcal{S}$ | $\mathcal{A}^\mathcal{T}$ | $\mathcal{H}$ | $\mathcal{A}^\mathcal{S}$ | $\mathcal{A}^\mathcal{T}$ | $\mathcal{H}$ |
| Full FT | 47.04 | 66.68 | 55.16 | 48.20 | 102.07 | 65.48 |
| $M[v] = 1$, when $v$ satisfy [Metric] | | | | | | |
| $\text{Rand}_\gamma$ | 51.68 | 70.15 | 59.52 | 50.16 | 106.18 | 68.13 |
| $\mathcal{I}_\gamma$ Eq. (2c) | 52.47 | 70.28 | 60.08 | 51.07 | 106.10 | 68.95 |
| $\mathcal{G}_\gamma$ Eq. (4) | 47.02 | 67.60 | 55.46 | 47.99 | 105.33 | 65.93 |
| $\mathcal{G}[v] > \mathcal{I}[v]$ Eq. (5) | **52.68** | **73.43** | **61.35** | **51.70** | **111.29** | **70.60** |

Table 4: **Ablative Study of Key Modules** for SPIDER. Incorporate **sole** Importance Discrepancy Measurement (IDM) can be regarded as Eq. (5). Considering **both** IDM and ISM, this is viewed as Eq. (8). For a detailed discussion, please refer to Sec. 4.2.

| IDM | ISM | Flickr30k | | | COCO-Capation | | |
|---|---|---|---|---|---|---|---|
| | | $\mathcal{A}^\mathcal{S}$ | $\mathcal{A}^\mathcal{T}$ | $\mathcal{H}$ | $\mathcal{A}^\mathcal{S}$ | $\mathcal{A}^\mathcal{T}$ | $\mathcal{H}$ |
| Zero-shot | | 61.39 | 55.43 | 58.26 | 61.39 | 107.64 | 78.19 |
| Full FT | | 47.04 | 66.68 | 55.16 | 48.20 | 102.07 | 65.48 |
| ✓ | | 52.68 | 73.43 | 61.35 | 51.70 | 111.29 | 70.60 |
| ✓ | ✓ | **55.40** | **83.49** | **66.61** | **55.94** | **122.74** | **76.86** |

specialized behavior. In Tab. 3, we examine the impact of different mask updating strategies. Specifically, weight magnitude and gradient value serve as two effective metrics for assessing parameter importance relative to the current distribution. $\mathcal{I}_\gamma$ and $\mathcal{G}_\gamma$ denote the selection of $\gamma$ proportion of elements based on small pre-trained weight magnitude and large downstream gradient value, respectively. Three key observations emerge: ❶ Selecting partial parameters is an effective solution for balancing generalization and specialization. ❷ Solely considering the pre-training distribution ($\mathcal{I}_\gamma$) emphasizes generalization-related elements but significantly limits the downstream adaptation. ❸ Exclusively incorporating fine-tuning importance ($\mathcal{G}_\gamma$) undermines generalization and hinders specialization. As demonstrated in Tab. 3, we select relative downstream important elements ($\mathcal{G}[v] > \mathcal{I}[v]$ in Eq. (5)), which effectively preserves generalization while ensuring specialization performance for Multimodal Large Language Model (MLLM).

**Key Component Analysis**. In Tab. 4, we begin by validating the significance of our proposed components through their incremental integration. The first row displays the BASELINE result, representing a simple Full Fine-Tuning (Full FT) approach using standard cross-entropy loss. As demonstrated, the combination of Importance Discrepancy Measurement (IDM) and Importance Selection Mask (ISM) yields the best performance in both generalization and specialization. This finding supports our motivation to evaluate parameter significance across pre-training and fine-tuning distributions, while selectively updating candidate parameters for the downstream task. Additionally, we plot the response outputs in Fig. 3, revealing that Zero-shot fails to adhere to the instruction style and lacks a detailed description. In contrast, naive Full FT introduces hallucinations,

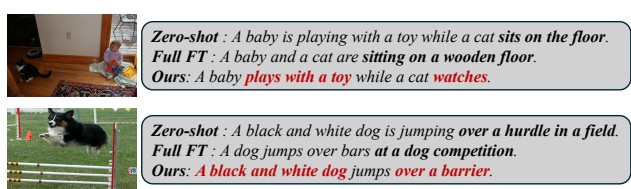

Figure 3: **Ablation Comparison on Response Output** on Flickr30k. Text prompt is *Write a short description for the image*. Full FT better follow the instructions than Zero-shot, but Full FT introduces hallucination (*e.g.*, "at a dog competition"), while Zero-shot lacks task details. Please refer to Sec. 4.2.

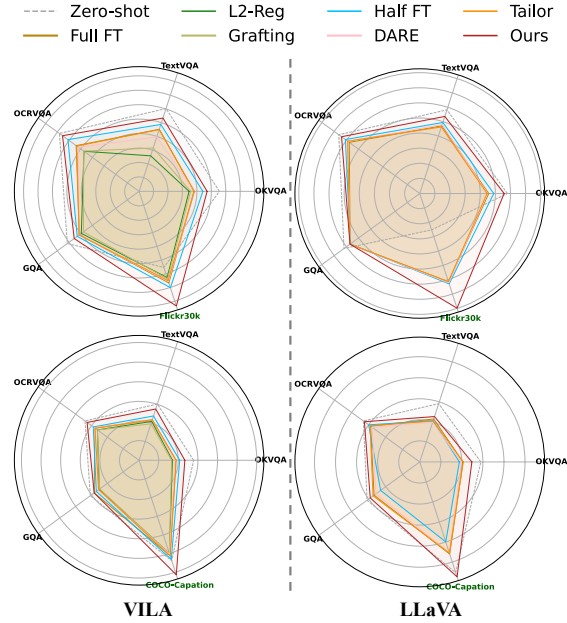

Figure 4: **Visualization Comparison**. Radar charts plots fine-tuning methods results across four pre-trained source datasets and target datasets, *i.e.*, Flickr30k and COCO-Capation. Our method achieves a better generalization and specialization trade-off.

likely due to the forgetting of generalized knowledge. Our method effectively achieves satisfying results.

### 4.3. Comparison to State-of-the-Arts

**Quantitative Results**. We compare our SPIDER against related approaches on image-captioning (CAP) and visual question-answering (VQA) tasks. Due to the architectural complexity and task differences, we limit the VQA evaluation to VILA1.5-3B. As shown in Tabs. 5 and 6, several key observations can be made: The larger the task gap between the fine-tuning and pre-training distributions, the more severe the generalization-specialization trade-offs in the MLLM fine-tuning process. Full-FT directly updates all parameters to adapt to the downstream distribution. However, specialized downstream applications typically have limited data compared to large-scale pre-training datasets. Thus, Full-FT often leads to increased overfitting, especially when training for more epochs, resulting in performance degradation on downstream tasks. Notably, regularization-

Table 5: **Comparison with the state-of-the-art Multimodal Large Language Model (MLLM) Fine-Tuning Solutions** on the image caption task: Flickr30k and COCO-Capation datasets based on VILA and LLaVA architectures. We mark the Best in bold across different tuning methods. ↑ means improved accuracy compared with Full FT. Please refer to Sec. 4.3 for relative explanations.

| Methods | Flickr30k | | | | | | | | COCO-Capation | | | | | | | |
|---|---|---|---|---|---|---|---|---|---|---|---|---|---|---|---|---|
| | OKVQA | TextVQA | OCRVQA | GQA | $\mathcal{A}^S$ | $\mathcal{A}^T$ | $\mathcal{H}$ | $\mathcal{O}$ | OKVQA | TextVQA | OCRVQA | GQA | $\mathcal{A}^S$ | $\mathcal{A}^T$ | $\mathcal{H}$ | $\mathcal{O}$ |
| *Fine-Tune with VILA architecture* | | | | | | | | | | | | | | | | |
| Zero-shot | 55.60 | 60.30 | 68.20 | 61.47 | 61.39 | 55.43 | 58.26 | 58.41 | 55.60 | 60.30 | 68.20 | 61.47 | 61.39 | 107.64 | 78.19 | 84.52 |
| Full FT | 37.99 | 45.17 | 53.85 | 51.14 | 47.04 | 66.68 | 55.16 | 56.86 | 37.36 | 42.96 | 55.85 | 56.63 | 48.20 | 102.07 | 65.48 | 75.14 |
| L2-Reg | 34.59 | 25.89 | 47.20 | 49.48 | 39.29 | 62.77 | 48.33 | 51.03 | 33.98 | 41.67 | 52.55 | 50.25 | 44.61 | 99.84 | 61.67 | 72.23 |
| Grafting | 35.66 | 31.60 | 47.40 | 47.67 | 40.58 | 63.71 | 49.58 | 52.15 | 33.12 | 39.06 | 52.60 | 49.77 | 43.64 | 99.84 | 60.73 | 71.74 |
| Half FT | 44.15 | 48.71 | 60.90 | 52.97 | 51.68 | 70.15 | 59.52 | 60.92 | 41.41 | 47.47 | 57.80 | 53.94 | 50.16 | 106.18 | 68.13 | 78.17 |
| DARE | 38.38 | 39.69 | 52.05 | 51.33 | 45.36 | 65.67 | 53.66 | 55.52 | 36.73 | 43.34 | 56.5 | 51.33 | 46.98 | 100.70 | 64.06 | 73.84 |
| Tailor | 38.30 | 44.98 | 53.35 | 51.38 | 47.00 | 65.00 | 54.56 | 56.00 | 37.84 | 43.51 | 55.70 | 50.96 | 47.00 | 102.44 | 64.44 | 74.72 |
| SPIDER | 47.11 | 53.38 | 65.55 | 55.57 | **55.40** | **83.49** | **66.61**↑11.45 | **69.45**↑12.59 | 46.65 | 54.94 | 65.55 | 56.63 | **55.94** | **122.74** | **76.86**↑11.38 | **89.34**↑14.20 |
| *Fine-Tune with LLaVA architecture* | | | | | | | | | | | | | | | | |
| Zero-shot | 58.00 | 58.25 | 66.20 | 61.93 | 61.10 | 25.31 | 35.79 | 43.20 | 58.00 | 58.25 | 66.20 | 61.93 | 61.10 | 110.52 | 78.69 | 85.81 |
| Full FT | 45.59 | 47.09 | 57.65 | 56.94 | 51.82 | 61.58 | 56.28 | 56.70 | 41.01 | 42.89 | 57.75 | 53.67 | 48.83 | 92.01 | 63.80 | 70.42 |
| Half FT | 48.96 | 49.47 | 60.80 | 56.81 | 54.01 | 62.91 | 58.12 | 58.46 | 37.62 | 41.14 | 60.00 | 45.96 | 46.18 | 79.91 | 58.53 | 63.05 |
| DARE | 44.82 | 48.01 | 58.75 | 57.04 | 52.16 | 62.18 | 56.73 | 57.17 | 39.60 | 39.45 | 56.00 | 51.50 | 46.64 | 90.82 | 61.63 | 68.73 |
| Tailor | 44.50 | 46.32 | 59.00 | 57.14 | 51.74 | 61.27 | 56.10 | 56.51 | 41.35 | 40.85 | 58.45 | 54.87 | 48.88 | 90.94 | 63.58 | 69.91 |
| SPIDER | 55.81 | 53.67 | 63.95 | 57.04 | **57.62** | **79.84** | **66.93**↑10.65 | **68.73**↑12.03 | 49.50 | 45.33 | 65.00 | 58.14 | **54.49** | **114.74** | **73.89**↑10.09 | **84.62**↑14.20 |

Table 6: **Comparison with the state-of-the-art Multimodal Large Language Model (MLLM) Fine-Tuning Solutions** on the the visual question answering task: IconQA and ScienceQA datasets based on the VILA architecture. Please see details in Sec. 4.3.

| Methods | IconQA | | | | | | | | ScienceQA | | | | | | | |
|---|---|---|---|---|---|---|---|---|---|---|---|---|---|---|---|---|
| | OKVQA | TextVQA | OCRVQA | GQA | $\mathcal{A}^S$ | $\mathcal{A}^T$ | $\mathcal{H}$ | $\mathcal{O}$ | OKVQA | TextVQA | OCRVQA | GQA | $\mathcal{A}^S$ | $\mathcal{A}^T$ | $\mathcal{H}$ | $\mathcal{O}$ |
| Zero-shot | 55.60 | 60.30 | 68.20 | 61.47 | 61.39 | 19.93 | 30.09 | 40.66 | 55.60 | 60.30 | 68.20 | 61.47 | 61.39 | 69.89 | 65.37 | 65.64 |
| Full FT | 34.51 | 38.02 | 46.10 | 47.05 | 41.42 | 87.05 | 56.13 | 64.24 | 47.15 | 50.88 | 57.20 | 53.58 | 52.20 | 75.78 | 61.82 | 63.99 |
| L2-Reg | 21.69 | 25.89 | 35.20 | 37.09 | 29.97 | 86.40 | 44.50 | 58.18 | 43.65 | 48.13 | 51.80 | 50.42 | 48.50 | 76.40 | 59.33 | 62.45 |
| Grafting | 22.66 | 31.60 | 40.25 | 37.84 | 33.09 | 87.18 | 47.97 | 60.13 | 45.65 | 50.71 | 54.35 | 53.94 | 51.16 | 76.00 | 61.16 | 63.58 |
| Half FT | 43.36 | 48.71 | 55.25 | 53.03 | 50.09 | **88.19** | 63.89 | 69.14 | 52.07 | 54.47 | 60.55 | 57.52 | 56.15 | 76.77 | 64.86 | 66.46 |
| DARE | 36.88 | 39.69 | 45.15 | 48.09 | 42.45 | 88.11 | 57.30 | 65.28 | 47.61 | 50.39 | 57.55 | 55.08 | 52.66 | **77.46** | 62.69 | 65.06 |
| Tailor | 37.99 | 41.42 | 48.10 | 47.57 | 43.77 | 88.16 | 58.50 | 65.97 | 48.22 | 50.98 | 57.90 | 53.04 | 52.54 | 75.97 | 62.12 | 64.25 |
| SPIDER | 48.34 | 53.38 | 63.35 | 56.12 | **55.30** | 84.07 | **66.71**↑10.58 | **69.68**↑5.44 | 54.13 | 57.33 | 65.60 | 60.07 | **59.28** | 75.97 | **66.60**↑4.78 | **67.63**↑3.64 |

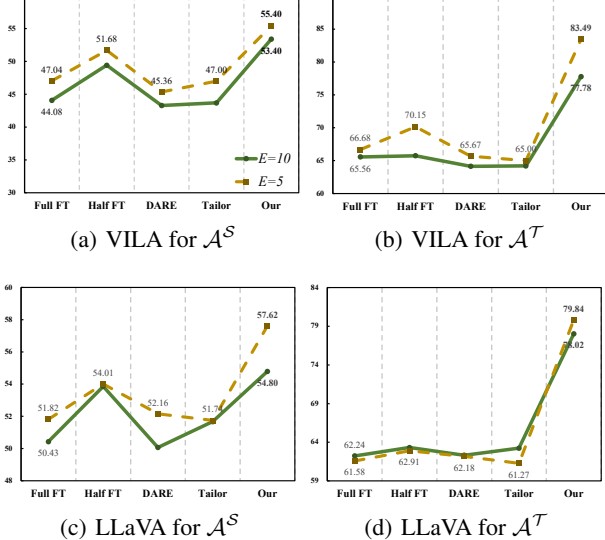

(a) VILA for $\mathcal{A}^S$    (b) VILA for $\mathcal{A}^T$

(c) LLaVA for $\mathcal{A}^S$    (d) LLaVA for $\mathcal{A}^T$

Figure 5: **Comparison on Large Fine-Tuning Epochs.** $E$ varies from (5 rounds to 10 rounds) on Flickr30k. See Sec. 4.3.

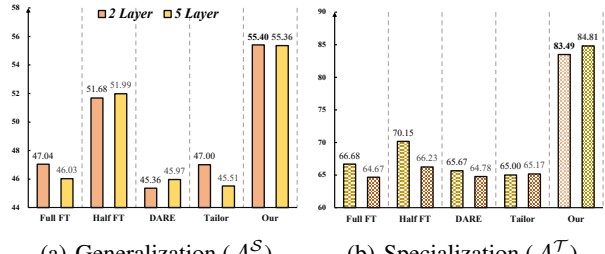

(a) Generalization ($\mathcal{A}^S$)    (b) Specialization ($\mathcal{A}^T$)

Figure 6: **Comparison on More Fine-Tuning Layer** $L$ on Flickr30k dataset with the VILA architecture. See Sec. 4.3.

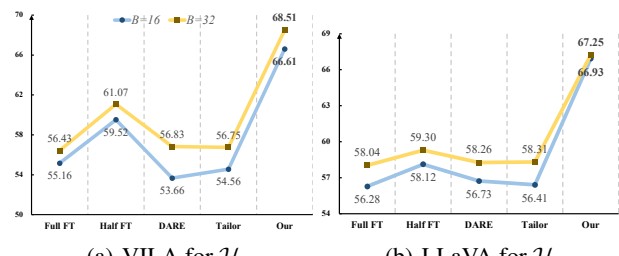

(a) VILA for $\mathcal{H}$    (b) LLaVA for $\mathcal{H}$

Figure 7: **Comparison on Large Training Batch Size** $B$ on Flickr30k dataset with VILA and LLaVA. Refer to Sec. 4.3.

based optimization approaches typically offer limited performance improvements, as controlling parameter stiffness to regulate the extent of LLM updates remains a challenging task. Moreover, in partial-update methodologies, directly combining updated parameters with pre-trained ones often leads to performance fluctuation, largely due to the influence of delta parameters scale. In contrast, both Half FT and our proposed method guide the LLM to fine-tune on selected parameters, demonstrating competitive performance

across various experiments. Additionally, our approach selectively targets relatively important parameters for downstream tasks, yielding better performance compared to the random selection strategy employed in Half FT. We further plot the radar visualization in Fig. 4 to highlight the performance advantages of ours compared to other approaches.

**Performance on More Tuning Epochs** $E$. We investigate

the impact of extending fine-tuning epochs $E$ from 5 to 10 rounds, shown in Fig. 5. The results highlight several key findings: (i) Extending fine-tuning epochs intensifies the pre-training knowledge forgetting phenomenon across different architecture scales. (ii) Smaller architectures, such as VILA-1.5-3B, encounter more severe parameter conflicts, where a decline in generalization ability results in degraded specialization performance. (iii) Larger architectures, such as LLaVA-1.5-7B, which possess higher parameter redundancy, maintain more stable specialization ability despite extended tuning epochs. (iv) Ours achieves robust performance across various architectures and training duration.

**Performance on More LLM Tuning Layer** $L$. We evaluate the effect of tuning block layers $L$ from 2 to 5, as shown in Fig. 6. Existing methods show a slight improvement in general performance but significantly reduce target domain performance. SPIDER effectively maintains both generalization and specialization across various tuning layers.

**Performance on Large Training Batch** $B$. We further conduct the experiments on the large training batch $B$: 32 in Fig. 7. The results show that setting a higher training batch benefits both generalization and specialization ability across different counterparts. Notably, our method SPIDER constantly achieves the best performance results.

## 5. Conclusion

In conclusion, we address the catastrophic forgetting in fine-tuning Multimodal Large Language Model (MLLM). We introduce Specialization via Importance Discrepancy Evaluation for Refinement (SPIDER 😡), a novel approach to assess parameter importance for both generalization and specialization, focusing on identifying downstream-important elements and performing critical-aware updates on selected parameters. Our method enjoys third advantages: First, *No Architecture Dependency*: SPIDER functions without specific model architecture, which presents high transferability across different architectures. Second, *No Fine-tuning Pattern Conflict*: we conduct partial parameter updates, maintaining compatibility with various optimization functions. Third, *No Hyper-Parameter Configuration*: leveraging parameter importance discrepancies requires no additional hyper-parameters, enhancing fine-tuning effectiveness. SPIDER has been validated on fruitful scenarios, highlighting the potential for broader applications.

## Acknowledgement

This work is supported by National Natural Science Foundation of China under Grant (62225113, 62361166629, 62176188, 623B2080), the National Key Research and Development Program of China (2023YFC2705700, 2024YFC3308400), and the Wuhan University Undergraduate Innovation Research Fund Project. The supercomputing system at the Supercomputing Center of Wuhan University supported the numerical calculations in this paper. Dr Tao's research is supported by NTU RSR and Start Up Grants. Besides, we thank Zekun Shi for his valuable collaboration. This work would not have been possible without him.

## Impact Statement

This paper presents work whose goal is to advance the field of Machine Learning. There are many potential societal consequences of our work, none of which we feel must be specifically highlighted here.

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

# APPENDIX

## A. Algorithm

We provide the algorithm description in Algorithm 1. The detailed method description is in Sec. 3.2.

---

**Algorithm 1** SPIDER

---

**Input:** Fine-Tuning Epoch $E$, Overall MLLM Network $\theta$, Trainable parameter module $w$, Frozen Pre-trained parameter weight $w^*$,

**Output:** The optimized selected MLLM model $w$

/* Generalization Knowledge Rank */
$\mathcal{I} \leftarrow (w^*)$ in Eq. (2c)
**for** $e = 1, 2, ..., E$ **do**
    **for** $(x^v, x^t, y) \in D^k$ **do**
        $h^v = \varphi(f(x^v)), h^t = \text{Tokenize}(x^t)$
        $\delta = \nabla \mathcal{L}\left(g([h^v, h^t]), y\right)$ via Eq. (3)
        /* Specialization Knowledge Rank */
        $\mathcal{G} \leftarrow (\delta)$ in Eq. (4)
        /* Importance Selection Mask */
$$M[v] = \begin{cases} \frac{\mathcal{G}[v]}{\mathcal{G}[v] + \mathcal{I}[v]}, & \mathcal{G}[v] > \mathcal{I}[v], \\ 0, & \text{else.} \end{cases}$$
        $\downarrow$ **Rescale** Importance Mask Matrix $M$
$$\boxed{M[v]} = \begin{cases} \min\left(1, \frac{M[v]}{\text{Mean}(M[M \neq 0])}\right), & \mathcal{G}[v] > \mathcal{I}[v], \\ 0, & \text{else.} \end{cases}$$
        $w = w - \eta \nabla \mathcal{L}$ ;         // Update Param.
        $w = w \odot M + w^* \odot (1 - M)$ Eq. (6)
    **end**
**end**

---

## B. Compared Methods

We focus on exploring model-agnostic MLLM fine-tuning methods, with a primary comparison between Regularization-based Optimization and Partial-based Updating solutions, as outlined below:

- Full Fine-Tuning (Full FT) [arXiv'05] (De Boer et al., 2005): Default optimize full parameters towards the downstream task.

- L2-Regularization (L2-Reg) [PNAS'17] (Kirkpatrick et al., 2017): Add an $\mathcal{L}_2$ regularization term with the regularization hyper-parameter, *i.e.*, 1e-3, to the original loss function. Thus, it focuses on keeping the fine-tuning model closer to the pre-trained model, thereby mitigating forgetting.

- Grafting [ICML'23] (Panigrahi et al., 2023): Localize newly acquired skills inside fine-tuned language models, which could be regarded as $\mathcal{L}_1$ regularization with the penalty weigh, *i.e.*, 1e-6.

- Half Fine-Tuning (Half FT) [arXiv'24] (Hui et al., 2024): Randomly update half of the parameter blocks within each transformer layer at each iteration while freezing the other elements.

- DARE [ICML'24] (Yu et al., 2024): Parameters from the fine-tuned model are randomly selected and re-scaled to maintain ability on generalization and specialization aspects.

- Tailor [ICML'24] (Zhu et al., 2024): Preserve pre-trained parameters while replacing a small ratio of fine-tuned parameters, *i.e.*, 10 %, based on the salience and sensitivity analysis.

