# OpenReview forum: "Learn from Downstream and Be Yourself in Multimodal Large Language Models Fine-Tuning"
_ICML.cc/2025/Conference — ICML 2025 poster_

### Official Review · Reviewer_mGiB · 2025-03-03

**Overall Recommendation:** 4

**Summary:**

This paper addresses the issue of catastrophic forgetting in the fine-tuning of multimodal large language models (MLLMs). The proposed method, Specialization via Importance Discrepancy Evaluation for Refinement (SPIDER), introduces an innovative approach for measuring parameter importance by utilizing the magnitude of frozen pre-trained weights and fine-tuning gradients. Based on this, SPIDER applies an importance-aware weight allocation strategy to prevent the pre-trained knowledge forgetting during fine-tuning. Experimental results demonstrate that SPIDER enhances downstream specialization performance while effectively mitigating generalization degradation.

**Claims And Evidence:**

Yes.

**Essential References Not Discussed:**

NaN

**Experimental Designs Or Analyses:**

The experimental results are comprehensive. The author conducts various experiments with different relative method in various downstream tasks.

**Methods And Evaluation Criteria:**

Yes. The methods are well evaluated.

**Other Comments Or Suggestions:**

Typos: “COCO-Capation” should be “COCO-Caption” in L124, L439, etc.

**Other Strengths And Weaknesses:**

Pros:

The SPIDER method presented in this paper is novel in that it does not rely on additional parameters. Instead, it utilizes gradients to guide parameter updates, influencing the update direction and making the model’s forgetting mechanism interpretable. Experimental results validate its effectiveness on LLaVA and VILA. Moreover, the approach is compatible with other optimization strategies, offering both simplicity and efficiency.

Cons:
- While SPIDER focuses on mitigating forgetting, its memory complexity may pose challenges for scaling. It is better to discuss the memory cost with the existing methods for better understanding.
- The authors could plot the parameter modification scale with existing methods. For example, both dare and tailor select a partial element for updates. Besides, the notation table is also important to make the paper reading friendly.
- The text in Table 4 is too small, and some parts are obscured. Please consider increasing the font size or adjusting the layout for better readability.
- The meaning of “Architecture Couple” in Table 1 seems unclear. Please provide a more detailed explanation of the significance of SPIDER and its advantages in the context of “Architecture Couple”.
- Why OKVQA, TextVQA, GQA, and OCRVQA are chosen as the datasets to evaluate the generalization ability? What do these datasets represent, and how do they contribute to assessing generalization?

**Questions For Authors:**

-Please refer to Other Strengths And Weaknesses.

**Relation To Broader Scientific Literature:**

The author introduces a novel aspect to alleviate the forgetting during MLLM fine-tuning via both pre-trained magnitude and current gradient information.

**Theoretical Claims:**

The authors provide clear normalization boundary analysis for both the weight and gradient parts.

---

> ### Author Rebuttal · Authors · 2025-03-26
>
> Dear Reviewer mGiB:
>
> Thank you for your valuable comments and constructive feedback. Below, we carefully address each of your concerns point-by-point, providing detailed explanations and additional evidence to clarify our approach.
>
> **Q1: Memory Complexity** (Other Strengths And Weaknesses)
>
> A1: For our method, we utilize the pre-trained weights and the current gradient to evaluate parameter importance for both generalization and specialization, which brings a certain memory cost increase with existing methods. However, our method achieves stable performance across different tuning scenarios, as shown in the following Table. We will add the memory complexity comparison in the final version.
>
> **Q2: Parameter Modification Scale** (Other Strengths And Weaknesses)
>
> A2: Sorry for the misunderstanding. We plot the parameter modification scale in the following Table. We will add the parameter modification scale and notation table in the final version.
>
> *Table: Comparison with relative methods. $M$ means selected parameters. Performance is the harmonic mean of generalization and specialization accuracy.*
> |Method |Full FT|L2-Reg|Grafting|Half FT|DARE|Tailor|Ours|
> |----|---|---|---|---|---|---|--|
> |Memory Complexity|$\mathcal{O}(\|M\|)$|$\mathcal{O}(2\|M\|)$|$\mathcal{O}(2\|M\|)$|$\mathcal{O}(\|M\|)$|$\mathcal{O}(\|M\|)$|$\mathcal{O}(\|M\|)$|$\mathcal{O}(3\|M\|)$|
> |Learnable Ratio|100%|100%|100%|50%|10%|10%|50%
> |Performance (VILA on Flickr30k)|55.16|48.33|49.58|59.52|53.66|54.56| **66.61**
>
> **Q3: Font size and Layout** (Other Strengths And Weaknesses)
>
> A3: Thanks for your advice. We will adjust the font size and layout to make it more readable!
>
> **Q4: Advantage Discussion for Proposed Method** (Other Strengths And Weaknesses)
>
> A4: Our method selects and optimizes the relative downstream important elements to adapt towards the downstream specialization while maintaining the generalization ability. Thus, our method does not bring additional architecture modification and does not couple with a specific model architecture. We will update Table 1 and provide a more detailed advantages discussion for our method in the final version.
>
> **Q5: Explanation and Contribution for Selected Datasets** (Other Strengths And Weaknesses)
>
> A5: The datasets OKVQA, TextVQA, GQA, and OCRVQA are obviously mentioned as the training datasets in the pre-training stage, making them appropriate benchmarks to evaluate multimodal large language models’ (MLLMs) generalization across diverse tasks. OKVQA examines external knowledge and common-sense reasoning, TextVQA and OCRVQA test understanding of embedded textual information, and GQA assesses compositional and logical reasoning. Together, these datasets comprehensively evaluate MLLMs’ generalization across different reasoning types and practical scenarios. We will provide a detailed explanation of the selected datasets in the paper. Thanks for your comments!
>
> **Q6: Typos** (Other Comments Or Suggestions)
>
> A6: Thanks for your advice. We will fix the typos and check the final version carefully.

---

> > ### Comment · Reviewer_mGiB · 2025-04-03
> >
> > After reading the rebuttal, the authors have adequately addressed my concerns. I increase my score.

---

### Official Review · Reviewer_CY1z · 2025-03-12

**Overall Recommendation:** 4

**Summary:**

In this work, it focuses on the Multimodal Large Language Models (MLLMs) fine-tuning field and reveals the catastrophic forgetting on the pre-training knowledge. Authors assess parameter importance for both generalization and specialization, focusing on identifying downstream-important elements and performing critical-aware updates on selected parameters. Through a series of experiments, SPIDER is shown to effectively enhance task-specific performance, such as in image captioning and visual question answering, while preventing the degradation of generalization. This work offers an insightful contribution to improving fine-tuning practices for large, pre-trained multimodal models.

**Claims And Evidence:**

The claim made in the paper is clear.

**Essential References Not Discussed:**

It is encouraged to add some discussions with incremental learning, as catastrophic forgetting is also a challenging and popular problem in incremental learning.

**Experimental Designs Or Analyses:**

The experimental results are comprehensive. The author conducts various experiments with different relative methods in various downstream tasks.

**Methods And Evaluation Criteria:**

The authors respectively measure the downstream and upstream performances and further utilize the upstream-downstream mean value to measure the overall performance.

**Other Comments Or Suggestions:**

The paper is written in a relatively complex manner, which may make it difficult for readers who are not experts in this specific field to fully understand it. It would benefit from improvements in clarity and accessibility, making it easier for a broader audience to grasp the key ideas and contributions.

**Other Strengths And Weaknesses:**

Strengths:
The approach of leveraging parameter magnitude and gradient norm to identify key task-specific parameters is compelling. The authors conduct extensive experiments across multiple datasets, including comprehensive ablation studies, to validate their method.

Weaknesses:
Given that SPIDER focuses on fine-tuning Multimodal Large Language Models (MLLMs), the authors should report both generalization and forgetting metrics, as well as specialization improvements, to provide a more thorough comparison with existing methods.

As for the downstream dataset selection, the author should plot the detailed data introduction for their response type and which field they belong to.

**Questions For Authors:**

The authors should add more discussion with incremental learning as catastrophic forgetting is a popular research problem in that scope. Besides, relative forgetting and increase metrics are also curial for readers to understand your paper.

**Relation To Broader Scientific Literature:**

The method is interesting as it achieves better downstream performance via measuring the parameter importance discrepancy.

**Theoretical Claims:**

NaN

---

> ### Author Rebuttal · Authors · 2025-03-26
>
> Dear Reviewer CY1z:
>
> Thank you for your insightful review and valuable feedback. Below, we address your key concerns in detail, aiming to clarify and demonstrate the effectiveness of our proposed approach.
>
>
> **Q1: Discussion on Incremental Learning** (Essential References Not Discussed & Questions For Authors)
>
> A1: Incremental learning focuses on continuously learning new tasks that are distinct from previously learned tasks, with the requirement to classify all observed classes during the testing phase [1,2,3]. Direct optimization solely on the current distribution can result in catastrophic forgetting of previously acquired knowledge. Existing incremental learning methods broadly fall into three categories: parameter isolation methods [4,5], regularization-based methods [6,7], and replay-based methods [8,9]. In this work, we investigate catastrophic forgetting within the multimodal large language model (MLLM) tuning process, aiming to mitigate the degradation of generalization and enhance specialization performance.
>
> [1] Class-incremental learning: A survey, IEEE PAMI, 2024
>
> [2] A continual learning survey: Defying forgetting in classification tasks, IEEE PAMI, 2021
>
> [3] A comprehensive survey of continual learning: theory, method and application, IEEE PAMI, 2024
>
> [4] Progressive Neural Networks, arXiv, 2016
>
> [5] Dense network expansion for class incremental learning, CVPR, 2023
>
> [6] Overcoming Catastrophic Forgetting in Neural Networks, PNAS, 2017
>
> [7] Memory aware synapses: Learning what (not) to forget, ECCV, 2018
>
> [8] icarl: Incremental classifier and representation learning, CVPR, 2017
>
> [9] Co-transport for class-incremental learning, ACM MM, 2021
>
> **Q2: Downstream Dataset Introduction** (Other Strengths And Weaknesses)
>
> For downstream evaluation, we selected datasets (**Flickr30k**, **COCO-Caption**, **IconQA**, **ScienceQA**) that exhibit limited zero-shot performance or were excluded from the pre-training stage. Flickr30k and COCO-Caption focus on text generation tasks within general domains. Specifically, **Flickr30k** emphasizes everyday activities and events, while **COCO-Caption** features more complex, diverse scenes involving multiple objects and intricate interactions. Both IconQA and ScienceQA involve multiple-choice question answering tasks. **IconQA** evaluates abstract and symbolic reasoning using visual questions based on icons and diagrams, whereas **ScienceQA** assesses multimodal scientific reasoning in educational contexts, incorporating textual and visual content from disciplines such as physics, chemistry, biology, and general science. We will incorporate a detailed introduction to these downstream datasets in our revised paper.
>
> **Q3: Writing Clarity and Accessibility** (Other Comments Or Suggestions & Questions For Authors)
>
> A3: Thank you for highlighting this. We will simplify the formulations and enhance the clarity of our writing in the revised manuscript.
>
> **Q4: Relative forgetting and increase metrics** (Questions For Authors & Other Strengths And Weaknesses)
>
> A4: Thank you for the valuable suggestion. We will incorporate relative forgetting and increase metrics into Tables 4 and 5 of the revised manuscript, explicitly reporting both generalization forgetting and specialization improvement to provide a clearer evaluation of method effectiveness.
>
> Thank you once again for your constructive feedback and valuable suggestions！

---

> > ### Comment · Reviewer_CY1z · 2025-04-03
> >
> > Thanks for the detailed clarifications and discussions. After reading the comments from other reviewers, I believe this is a good paper for the community and I would like to keep my positive rating.

---

### Official Review · Reviewer_mmMS · 2025-03-14

**Overall Recommendation:** 3

**Summary:**

This paper introduces a novel and well-structured strategy to address the persistent challenge of catastrophic forgetting that arises during the fine-tuning of Multimodal Large Language Models (MLLMs). This method leverages a meticulous analysis of parameter importance discrepancies to guide the optimization process, ensuring that previously learned knowledge is effectively retained while allowing the model to adapt and specialize for downstream tasks. By carefully balancing the trade-off between generalization and specialization, this approach enhances the model’s ability to perform robustly across diverse scenarios without compromising its adaptability to new domains. Furthermore, this strategy provides a systematic framework for mitigating knowledge degradation, ultimately improving the model’s stability and performance in real-world applications.

**Claims And Evidence:**

Yes. This paper follows the relative requirements.

**Essential References Not Discussed:**

It would be helpful if the authors included references related to large model merging, as they make comparisons with the Tailor and Dare methods.

**Experimental Designs Or Analyses:**

Although the authors conduct experiments on multimodal large language models tuning with larger training epochs, it would be better to provide the depth analysis why full-ft deceases the downstream performance with larger epochs.

**Methods And Evaluation Criteria:**

Yes. It includes difference metrics.

**Other Comments Or Suggestions:**

Please refer to the weaknesses.

**Other Strengths And Weaknesses:**

This paper explores the issue of catastrophic forgetting in Multimodal Large Language Models (MLLMs) fine-tuning. The manuscript is well-written, structured, and easy to follow, with the proposed method being clearly presented and demonstrating novelty. There are several weaknesses for updates.
-The authors should include a discussion of relative performance, particularly in comparison to Full-FT. Full-FT, while a widely adopted and straightforward fine-tuning solution, yields lower performance across various downstream tasks, as shown in the Table 4 and especially for the flickr30k and coco-caption. A more detailed analysis would provide a clearer context for evaluating propose method advantages.
-The authors should add a detailed discussion with the foundation model merge works. Because author compare the proposed method with both Tailor and Dare.

**Questions For Authors:**

If the authors address the issues mentioned in the weaknesses section, I would be happy to increase the score.

**Relation To Broader Scientific Literature:**

The author leverages both pre-training and downstream knowledge to construct an importance discrepancy mask, effectively mitigating forgetting while ensuring specialization. This approach presents an intriguing solution that enhances Multimodal Large Language Models (MLLMs) by striking a balance between domain-specific adaptation and generalization capability, ultimately improving their applicability and robustness in real-world scenarios.

**Theoretical Claims:**

The authors could specify the value range boundaries for sections 2a-2c and provide a rationale for selecting the sigmoid and normalization operations.

---

> ### Author Rebuttal · Authors · 2025-03-26
>
> Dear Reviewer mmMS:
>
> Thank you very much for your affirmation of our work, as well as the insightful concerns and questions you have raised. We have carefully considered each comment and provided responses.
>
> **Q1: Value Range Boundary and Normalization Operation Rationale** （Theoretical Claims）
> A1: Sections 2a–2c measure parameter importance for generalization by leveraging the pre-trained weight magnitudes. Specifically, step 2a computes the absolute values of pre-trained weights, yielding a range of [0,$+\infty$). Step 2b normalizes these absolute values to a standardized range of ($-\infty$,$+\infty$). Finally, step 2c rescales the normalized values into the range (0,1) using a sigmoid function. Together, these steps systematically rank parameters according to their importance for generalization, facilitating a meaningful comparison with parameters important for specialization. We will provide detailed explanations and discuss the rationale behind these operations in the revised manuscript. Thanks for your advice!
>
> **Q2: Depth Analysis on Full-FT** (Experimental Designs Or Analyses & Other Strengths And Weaknesses)
>
> A2: Full-FT directly updates all parameters to adapt to the downstream distribution. However, specialized downstream applications typically have limited data compared to large-scale pre-training datasets. Thus, Full-FT often leads to increased overfitting, especially when training for more epochs, resulting in performance degradation on downstream tasks. We will include a detailed analysis of Full-FT performance in the revised manuscript. Thank you for tips!
>
> **Q3: Differences between Regularization-based Optimization and Partial-based Updating solutions** (Supplementary Material)
>
> A3: Regularization-based optimization and partial-based updating represent two distinct approaches to addressing catastrophic forgetting in multimodal large language model (MLLM) tuning. The regularization-based approach introduces stiffness regularization terms, carefully balancing generalization and specialization through controlled penalization. In contrast, partial-based updating selects only a subset of parameters to update, preserving the rest. Our work adopts the partial-based updating paradigm, measuring parameter importance for both generalization and specialization, and selectively updating parameters significant for downstream tasks. We will clearly and concisely discuss the differences compared to existing methods in our revised manuscript. Thank you for your valuable suggestion!
>
> **Q4: Large Model Merging Paper Discussion** (Essential References Not Discussed & Other Strengths And Weaknesses)
>
> A4: Large Model Merging is a promising paradigm that integrates task-specific models into a unified model capable of simultaneously handling diverse downstream tasks [1]. Existing methods typically represent each task expert using downstream-optimized models and rely on various importance metrics to select specific elements for merging [2,3,4]. However, post-merging operations often disrupt the originally optimized parameter space, limiting the specialized capabilities for individual tasks. In our work, we select relative specialization importance elements during the training process, which is compatible with integrates into the training trajectory and squeeze the specialized capabilities into the target parameter space. We will add large model merging works discussion in our final version.
>
> [1] Learn From Model Beyond Fine-Tuning: A Survey, Nature Machine Intelligence, 2024
>
> [2] A Unified View of Delta Parameter Editing in Post-Trained Large-Scale Models, arXiv, 2024
>
> [3] Model Tailor: Mitigating Catastrophic Forgetting in Multi-modal Large Language Models, ICML, 2024
>
> [4] REMEDY: RECIPE MERGING DYNAMICS IN LARGE VISION-LANGUAGE MODELS, ICLR, 2025

---

### Official Review · Reviewer_mG7E · 2025-03-15

**Overall Recommendation:** 4

**Summary:**

This manuscript addresses the catastrophic forgetting issue in fine-tuning MLLMs. It proposes SPIDER, which measures parameter importance based on pre-trained weight magnitudes and fine - tuning gradients. SPIDER uses Importance Discrepancy Measurement (IDM) to rank parameter importance and Importance Selection Mask (ISM) for selective parameter updates. Experiments on image captioning and VQA with models like LLaVA and VILA show that SPIDER effectively balances generalization and specialization, outperforming baseline methods.

## update after rebuttal

The response has addressed all my concerns, and I will increase my score.

**Claims And Evidence:**

Overall, all claims are well-supported. However, one point that confuses me is the rationale behind simply comparing I[v] and G[v] after normalization and rescaling to determine which weights should be updated. This approach lacks sufficient justification. Specifically, could this method potentially lead to updates in nearly half of the weights? Given that general knowledge from pretraining typically outweighs specialized knowledge gained through fine-tuning, might this not be somewhat unreasonable or problematic?

**Essential References Not Discussed:**

The following paper is an important paper on overcoming the catastrophic forgetting of neural networks, but it is not cited or discussed in this paper.
[1] Aljundi R, Babiloni F, Elhoseiny M, et al. Memory aware synapses: Learning what (not) to forget[C]//Proceedings of the European conference on computer vision (ECCV). 2018: 139-154.

**Experimental Designs Or Analyses:**

No

**Methods And Evaluation Criteria:**

Yes

**Other Comments Or Suggestions:**

1. Consider how to improve the comparison between I[v] and G[v] so that the problem of generalization perturbation caused by fine-tuning under the same distribution can be solved. Specifically, in the case of close distribution, further dynamic reduction of update parameters may be difficult to achieve in the short term. 2. Adding missing traditional references and comparative experiments to overcome catastrophic forgetting.

**Other Strengths And Weaknesses:**

Strengths:
1. Novel solution: Proposes SPIDER to address catastrophic forgetting in MLLM fine-tuning by measuring parameter importance, offering a new way to balance generalization knowledge and specialization knowledge.
2. Comprehensive experiments: Tests on multiple MLLM architectures and datasets, comparing with SOTA methods. Ablation studies and various metrics validate its effectiveness.
3. Practical features: Architecture-agnostic, compatible with different optimization functions, and requires no hyper-parameter configuration.

Weaknesses
1. Memory issue: Ranking parameters by pre-trained weights and gradients increases memory usage, which may be a problem in resource-constrained settings.
2. Distribution sensitivity: The current approach determines parameter updates by simply comparing the normalized and rescaled values of I[v] and G[v], potentially leading to updates in nearly half of the parameters. Consequently, extensive parameter updates may occur, resulting in decreased generalization ability even when the downstream task distribution differs only slightly from the pre-trained distribution.
3. Lack of comparison with traditional methods: Catastrophic forgetting is not limited to the field of large models. There have been many studies on this aspect in neural networks, such as ref[1]. This article lacks a comparison with this traditional method.

[1] Aljundi R, Babiloni F, Elhoseiny M, et al. Memory aware synapses: Learning what (not) to forget[C]//Proceedings of the European conference on computer vision (ECCV). 2018: 139-154.

**Questions For Authors:**

See "Other comments or suggestions" for details.

**Relation To Broader Scientific Literature:**

Building upon the concept of catastrophic forgetting in neural networks, this paper proposes a novel approach to better balance generalized and specialized knowledge in MLLMs during fine-tuning, effectively mitigating forgetting.

**Theoretical Claims:**

The paper lacks rigorous mathematical proofs.

---

> ### Author Rebuttal · Authors · 2025-03-26
>
> Dear Reviewer mG7E:
>
> We sincerely appreciate your time and effort in reviewing our paper. Through the detailed responses outlined below, we seek to fully address your concerns and provide transparency into our proposed approach.
>
> **Q1: Essential Reference Discussion and Comparison** (Essential References Not Discussed & Other Strengths And Weaknesses & Other Comments Or Suggestions)
>
> A1: We apologize for overlooking the critical reference. Memory Aware Synapses (MAS) [1] addresses catastrophic forgetting in lifelong learning by measuring parameter importance through output sensitivity and applying a regularization penalty to prevent substantial updates to critical parameters. In our experiments, we selected OKVQA as a representative dataset to estimate parameter importance for generalization and compared MAS with different regularization hyperparameters ($\lambda$), as detailed in the table below. The results show that MAS outperforms other regularization-based methods; however, it provides limited trade-offs between generalization and specialization in the context of multimodal large language models (MLLMs). This limitation arises because the regularization terms cannot effectively maintain stable parameter constraints in large-scale models, negatively affecting overall performance. We will expand upon these observations and their implications in our revised manuscript.
>
> *Table: Performance comparison with relative methods. We conduct experiments on the VILA architecture and tune on the Flickr30k dataset.*
> |Method |OKVQA | TextVQA | OCRVQA | GQA | $\mathcal{A}^S$ | $\mathcal{A}^T$| $\mathcal{H}$| $\mathcal{O}$ |
> |----|---|---|---|---|---|---|--|--|
> | Zero-shot | 55.60 | 60.30 | 68.20 | 61.47 | 61.39 | 55.43 | 58.26 | 58.41|
> | Full FT   | 37.99 | 45.17 | 53.85 | 51.14 | 47.04 | 66.68 | 55.16 | 56.86|
> |L2-Reg| 34.59 | 25.89 | 47.20 | 49.48 | 39.29 |  62.77 | 48.33 | 51.03 |
> | MAS ($\lambda=0.01$)| 34.86 | 40.45 | 46.50 | 47.61 | 42.35 | 63.63 | 50.85 | 52.99
> |SPIDER (Ours)|47.11 | 53.38 | 65.55 |55.57 | **55.40** |**83.49** |**66.61**| **69.45**|
>
> [1] Memory aware synapses: Learning what (not) to forget, ECCV, 2018.
>
> **Q2: Distribution Sensitivity Effect on Parameter Importance Difference** (Essential References Not Discussed & Other Strengths And Weaknesses & Other Comments Or Suggestions)
>
> A2: Our method leverages pre-trained weights and current gradient updates to separately construct parameter importance rankings for generalization and specialization. In our first module, we identify elements relatively important for downstream tasks based on these rankings. Furthermore, we note that naively updating candidate parameters does not reflect parameter-level variance effectively (Eq. 6). Therefore, we propose the Importance Selection Mask, which dynamically allocates higher update rates to parameters exhibiting greater importance discrepancies, as defined in Eqs. 7-8 and illustrated on page 5. Consequently, our approach dynamically restricts parameter updates when the downstream task distribution closely aligns with the generalization distribution, effectively preserving generalization capability.

---

### Decision · Program_Chairs · 2025-05-01

**Decision:**

Accept (poster)

**Comment:**

This paper addresses catastrophic forgetting in fine-tuning Multimodal Large Language Models (MLLMs) by proposing SPIDER,  which balances generalization and specialization through parameter importance measurement. SPIDER employs frozen pre-trained weights and fine-tuning gradients to guide selective parameter updates. This work validates the approach across diverse architectures (e.g., LLaVA, VILA) and tasks (image captioning, VQA), demonstrating downstream performance improvement while mitigating generalization degradation.

This paper makes a timely contribution to MLLM fine-tuning by addressing catastrophic forgetting through a theoretically grounded and empirically validated approach. The reviewers have given uniformly positive ratings, ranging from 3 (weak accept) to 4 (accept). Based on the novelty, scalability, and comprehensive evaluation, the meta-review recommends acceptance of the submission.